# Uterine glands coordinate on-time embryo implantation and impact endometrial decidualization for pregnancy success

Andrew M. Kelleher[1], Jessica Milano-Foster[1], Susanta K. Behura[1] & Thomas E. Spencer [1,2]

Uterine glands are essential for pregnancy establishment. By employing forkhead box A2 (FOXA2)-deficient mouse models coupled with leukemia inhibitory factor (LIF) repletion, we reveal definitive roles of uterine glands in embryo implantation and stromal cell decidualization. Here we report that LIF from the uterine glands initiates embryo-uterine communication, leading to embryo attachment and stromal cell decidualization. Detailed histological and molecular analyses discovered that implantation crypt formation does not involve uterine glands, but removal of the luminal epithelium is delayed and subsequent decidualization fails in LIF-replaced glandless but not gland-containing FOXA2-deficient mice. Adverse ripple effects of those dysregulated events in the glandless uterus result in embryo resorption and pregnancy failure. These studies provide evidence that uterine glands synchronize embryo-endometrial interactions, coordinate on-time embryo implantation, and impact stromal cell decidualization, thereby ensuring embryo viability, placental growth, and pregnancy success.

---

[1] Division of Animal Sciences, University of Missouri, Columbia, 65211 MO, USA. [2] Department of Obstetrics, Gynecology and Women's Health, University of Missouri, Columbia, 65211 MO, USA. Correspondence and requests for materials should be addressed to T.E.S. (email: spencerte@missouri.edu)

Pregnancy establishment requires effective molecular cross-talk between a receptive uterus and an implantation competent embryo. In mice, blastocysts enter the uterus early on gestational day (GD) 4 (GD 1 is observation of a post-coital vaginal plug) and implantation commences within epithelial crypts formed on the antimesometrial side of the uterus around midnight on GD 4[1–4]. Embryo implantation involves blastocyst apposition, attachment, and adhesion to the luminal epithelium (LE)[5]. Decidualization of stromal cells commences on the morning of GD 5 near the attached blastocyst and eventually spreads toward the mesometrial area of the uterus[6]. Completion of the attachment reaction is evident with the removal of the LE by entosis, a cell-in-cell invasion phenomenon, during the night of GD 5[7]. By GD 6, the trophectoderm begins to directly contact the decidualizing stroma. In humans, asynchronous embryo-uterine interactions and defective stromal cell decidualization can result in pregnancy complications such as preeclampsia as well as pregnancy loss and miscarriage[3,5].

Uterine glands have established or postulated biological roles in the establishment of pregnancy in both mice and humans[8,9]. *Leukemia inhibitory factor* (*Lif*) is expressed during the window of receptivity and in mice is solely expressed by the glandular epithelium (GE) of the uterus on GD 4 in response to the nidatory surge in estrogen from the ovary. The infertility phenotype of *Lif* null mice as well as mice and sheep lacking uterine glands supports a primary role for gland-derived products in pregnancy establishment and maintenance[9–13]. Forkhead box (FOX) transcription factors regulate the development and function of many organs[14,15]. In the uterus of mice[10,11,13,16] and humans[17], FOXA2 is expressed explicitly in the glands. Genesis of endometrial glands in the neonatal uterus is compromised by conditional deletion of *Foxa2* using the progesterone receptor (Pgr)-Cre mouse model, which ablates genes in the endometrial epithelium, stroma and inner circular myometrium of the uterus after birth[10]. In contrast, glands are present in the adult uterus with conditional deletion of *Foxa2* using the lactotransferrin (Ltf)-iCre mouse model, as it ablates genes specifically in the LE and GE only after puberty[16]. Both FOXA2-deficient mouse models are infertile due to defects in embryo attachment and lack LIF expression on GD 4[10,16]. Embryo implantation can be rescued in both mouse models by intraperitoneal injections of LIF on GD 4. In glandless mice ($Pgr^{Cre/+}Foxa2^{f/f}$), pregnancies fail by GD 10, whereas they are maintained to term in gland-containing FOXA2-deficient mice ($Ltf^{iCre/+}Foxa2^{f/f}$). These studies strongly support the hypothesis that uterine glands and, by inference, their products have essential biological roles in embryo implantation and stromal cell decidualization for the establishment of pregnancy[18].

Perturbations in blastocyst positioning and timing of implantation elicit adverse ripple effects and compromise pregnancy outcomes in mice[3,19,20]. Here, a series of studies were conducted with FOXA2-deficient mouse models to understand the origin of pregnancy loss in LIF-replaced mice that lack uterine glands. The studies conclude that on-time implantation is disrupted in LIF-replaced glandless mice based on defects in active removal of the LE and stromal cell decidualization that manifest in pregnancy failure. Therefore, these studies provide original evidence that uterine glands and, by inference, LIF and other gland-derived factors synchronize on-time embryo implantation and impact stromal cell decidualization that are crucial for the establishment of pregnancy.

## Results

**Transcriptome alterations in FOXA2-deficient uteri on GD 4.** Acquisition of uterine receptivity on GD 4 requires dynamic changes in gene expression in the uterine epithelia as well as stroma[2,3]. Transcriptome profiling of GD 4 uteri revealed that 7940 (3930 increased and 4010 decreased) genes differed in glandless $Pgr^{Cre/+}Foxa2^{f/f}$ uteri and 3489 (1743 increased and 1746 decreased) genes differed in gland-containing $Ltf^{iCre/+}Foxa2^{f/f}$ uteri compared to control uteri (Fig. 1a, b and Supplementary Data 1 and 2). Of particular note, expression of *Lif* and several other established GE-specific genes (*Cxcl15, Prss29, Spink3, Ttr, Wfdc3*) was substantially reduced or absent in the uteri of both FOXA2-deficient mouse models. Of the differentially expressed genes, 2303 were unique to $Pgr^{Cre/+}Foxa2^{f/f}$ compared to control uteri, and 361 were unique to $Ltf^{iCre/+}Foxa2^{f/f}$ compared to control uteri (Fig. 1b). Integration with uterine epithelial-specific transcriptomic data from our previous study[21] determined that 137 GE-enriched genes (GE > LE, ANOVA; $P < 0.05$, >2-fold) were differentially expressed in the uteri of glandless $Pgr^{Cre/+}Foxa2^{f/f}$ compared to control and gland-containing $Ltf^{iCre/+}Foxa2^{f/f}$ mice (Fig. 1b and Supplementary Data 3). Functional analysis of those 137 genes found enrichment for gene ontology (GO) terms including cell motility, cell migration, extracellular matrix, and basement membrane (Supplementary Data 4).

Next, the FANTOM5 database[22] was used to determine ligands and receptors in the GD 4 transcriptome data. This analysis identified four GE-enriched genes (*Fn1, Hp, Lama2, Sema3c*) that encode ligands differentially expressed only in the uteri of glandless $Pgr^{Cre/+}Foxa2^{f/f}$ mice and have cognate receptors expressed in the uterus (Fig. 1c and Supplementary Table 1). *Fn1* expression was over three-fold lower in the uteri of glandless $Pgr^{Cre/+}Foxa2^{f/f}$ as compared to control mice, and expression of several FN1 receptors was also lower in glandless $Pgr^{Cre/+}Foxa2^{f/f}$ uteri (Supplementary Data 2 and Supplementary Table 1). Collectively, these analyses support the idea that the glands mediate endometrial synchrony by producing secreted factors that interact with other endometrial cell types during the peri-implantation period.

**Crypt formation is not altered in FOXA2-deficient mice.** Blastocysts enter the uterus on the afternoon of GD 4 and become positioned in the implantation crypts formed in the anti-mesometrial region of the uterus prior to attachment[1]. Initiation of blastocyst attachment and subsequent nidation involves dynamic changes in gene expression observed in both LE and stromal cells adjacent to the blastocyst[2–4,19,23,24]. Both glandless $Pgr^{Cre/+}Foxa2^{f/f}$ and gland-containing $Ltf^{iCre/+}Foxa2^{f/f}$ mice are infertile due to embryo implantation failure and lack *Lif* expression on GD 4[16].

To assess blastocyst positioning and initiation of implantation, we carefully examined implantation sites during early pregnancy. Hatched blastocysts were present within implantation crypts on the antimesometrial side of the uterus in both control and FOXA2-deficient mice at 2200 hours on GD 4 (Fig. 2a). However, *heparin-binding EGF-like growth factor* (*Hbegf*), an essential mediator of embryo-uterine interactions during implantation and marker of trophectoderm attachment[25], was not upregulated in LE cells adjacent to the blastocyst in FOXA2-deficient mice as found in control mice (Fig. 2a).

**LIF initiates embryo attachment in FOXA2-deficient mice.** Intraperitoneal injections of recombinant mouse LIF on GD 4 will initiate embryo implantation in FOXA2-deficient mice[16]. Although pregnancy is maintained to term in LIF-replaced gland-containing $Ltf^{iCre/+}Foxa2^{f/f}$ mice, embryos are resorbed and pregnancy is lost by GD 10 in LIF-replaced glandless $Pgr^{Cre/+}Foxa2^{f/f}$ mice[16]. Here, LIF repletion was performed to examine

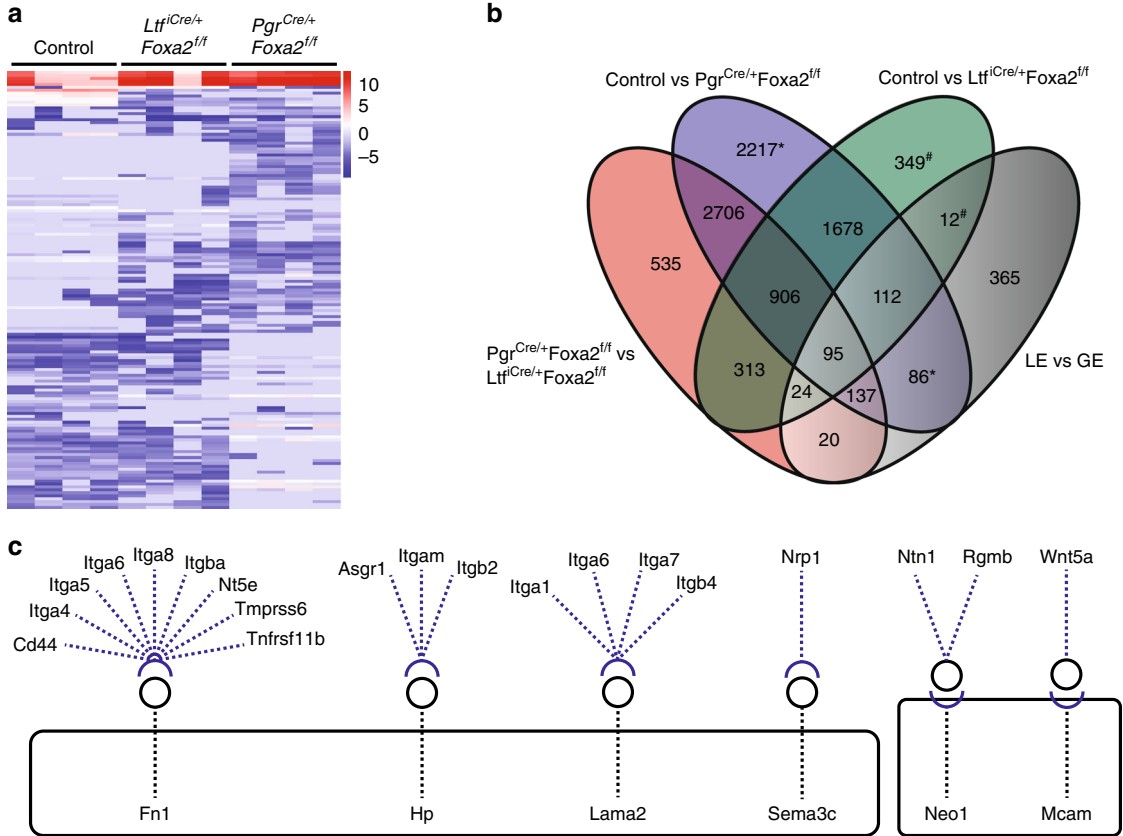

**Fig. 1** The uterine transcriptome is dysregulated in mice that lack glands. RNA-sequencing was performed using uteri from gland-containing *Ltf*^iCre/+ *Foxa2*^f/f glandless *Pgr*^Cre/+*Foxa2*^f/f, and control mice on GD 4. **a** Heatmap of the top 150 differentially expressed genes (log2 FPKM values) in FOXA2-deficient mice compared to controls. **b** Venn diagram comparing unique or common transcripts between the uterus of gland-containing *Ltf*^iCre/+*Foxa2*^f/f, glandless *Pgr*^Cre/+*Foxa2*^f/f, and control mice on GD 4. The LE vs GE represents genes that were enriched in the GE compared to the LE on GD 4. Superscripts denote genes combined for analysis. **c** Receptor and ligand interactions between genes uniquely differentially expressed in the uteri of glandless mice that are also enriched in GE on GD 4. Black boxes indicate genes uniquely differentially expressed in uteri of glandless *Pgr*^Cre/+*Foxa2*^f/f compared to control mice. Circles represent ligands, and half circles are receptors. All analysis was conducted using four biological replicates

embryo implantation in-depth. On the morning of GD 5, implantation sites were observed in both control and LIF-replaced FOXA2-deficient mice, whereas they were absent from uteri of FOXA2-deficient mice receiving saline vehicle on GD 4 (Fig. 2b). The number of implantation sites in the LIF-replaced FOXA2-deficient mice did not differ from control mice[16].

Next, *Hbegf* expression was determined by in situ hybridization, as *Hbegf* is upregulated in the LE cells adjacent to the implanting blastocyst beginning late on GD 4[26] (Fig. 2a, c). At 0900 hours on GD 5, *Hbegf* mRNA was present in the LE cells surrounding the embryo in the implantation chamber of control but not saline-treated FOXA2-deficient mice (Fig. 2c). Similar to control mice, *Hbegf* was clearly present in the LE of both types of FOXA2-deficient mice receiving LIF injections on GD 4 (Fig. 2c). The morphology of the implantation chambers formed in uteri of saline-treated FOXA2-deficient mice was consistently more spherical or ovoid in shape as compared to the spear shape of chambers formed in control mice.

On GD 4, GE cells respond to the nidatory surge in estrogen from the ovary and express *Lif* that activates signal transducer and activator of transcription 3 (STAT3) in the LE[12,27,28]. At 0900 hours on GD 5, activated phosphorylated STAT3 (pSTAT3) was present in the nuclei of LE and stromal cells adjacent to embryos in the implantation chamber of control but not saline-treated FOXA2-deficient mice (Supplementary Figure 1). Intra-peritoneal injections of LIF on GD 4 activated STAT3 in the LE

and stromal cells adjacent to the embryo in both types of FOXA2-deficient mice.

Expression of prostaglandin-endoperoxide synthase 2 (PTGS2) commences on the morning of GD 5 in the decidualizing stromal cells that form the primary decidual zone (Pdz) adjacent to the implanting blastocyst[24]. All implantation chambers of GD 5 control mice contained PTGS2-positive stromal cells near the implanting embryo, whereas PTGS2 was restricted to only LE cells juxtaposed to the embryo in uteri of GD 5 FOXA2-deficient mice (Supplementary Figure 2a). After LIF repletion on GD 4, implantation chambers in FOXA2-deficient mice contained PTGS2-positive stromal cells similar to control mice on the morning of GD 5 (Fig. 3a). Thus, LIF replacement initiates blastocyst implantation and onset of stromal cell decidualization in the uteri of both gland-containing and glandless FOXA2-deficient mice.

Epithelial and tight junction remodeling occurs in the uterus during implantation that is essential for pregnancy establishment and success[19,20,29,30]. We evaluated E-cadherin (CDH1) and claudin 1 (CLDN1) in the LE, as they are components of adherens junctions and tight junctions, respectively. At 0900 hours on GD 5, CDH1 was uniformly abundant in the LE cells of implantation crypts in both control and LIF-replaced FOXA2-deficient mice (Fig. 3b). The tight junction protein CLDN1 exhibited a more variegated pattern in the LE of implantation sites from control and LIF-replaced *Ltf*^iCre/+*Foxa2*^f/f mice, particularly in antime-sometrial region, as compared to LIF-replaced *Pgr*^Cre/+*Foxa2*^f/f

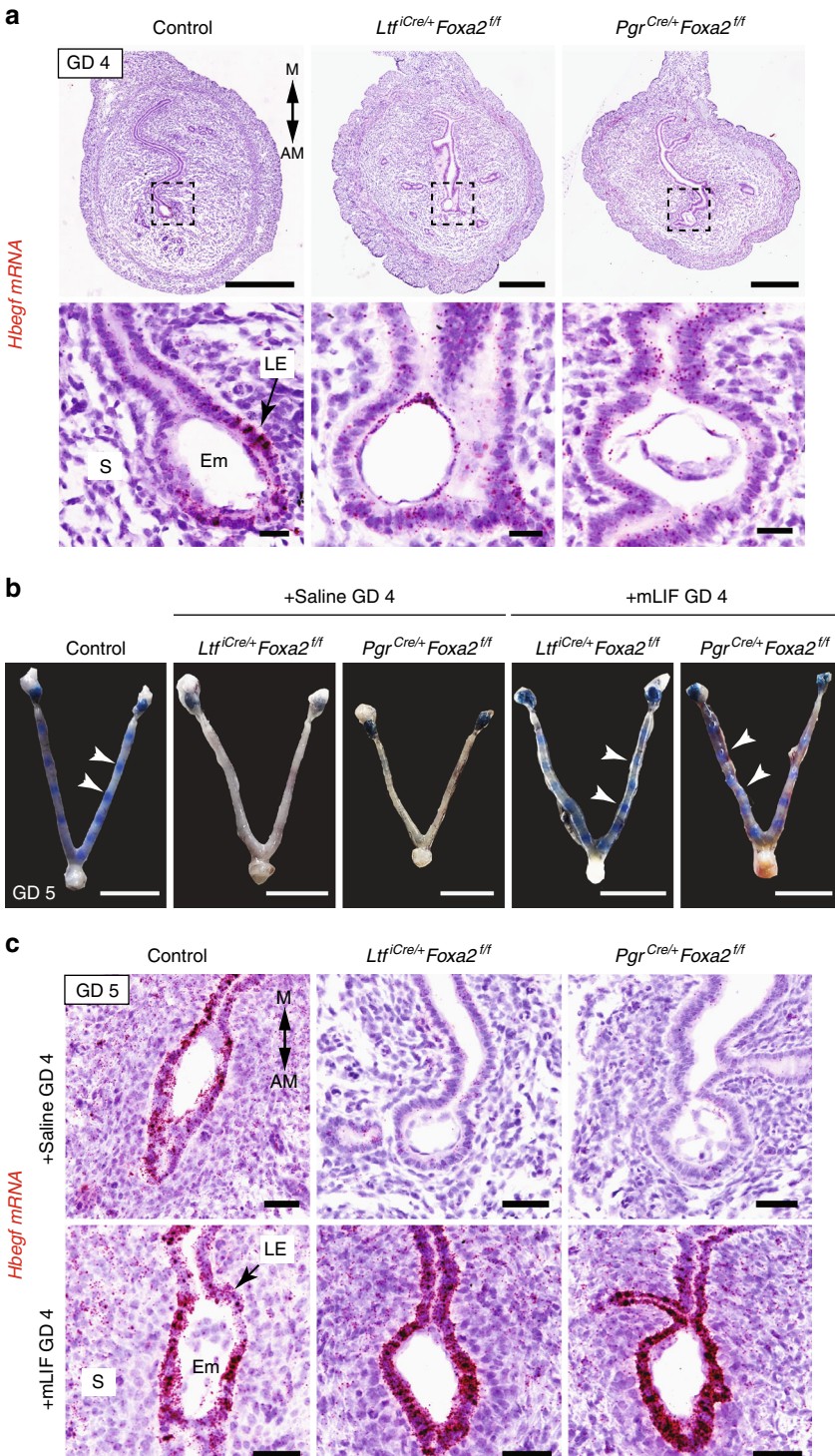

**Fig. 2** LIF initiates implantation in FOXA2-deficient mice. **a** In situ localization of *Hbegf* mRNA in the uterus of control and FOXA2-deficient mice without LIF repletion on GD 4 at 2200 hours. Uterine sections were counterstained with hematoxylin after chromogenic detection of *Hbegf* mRNA (red). Top panel— Scale bar: 500 μm; Bottom panel—Scale bars: 25 μm. **b** Gross morphology of uteri on GD 5 at 0800 hours. FOXA2-deficient mice received intraperitoneal (i.p.) injections of saline or recombinant mouse LIF on GD 4. Implantation sites accumulate Evans Blue Dye. White arrowheads point to individual implantation sites. Scale bar: 1 cm. **c** In situ localization of *Hbegf* mRNA in the uterus on GD 5 at 0800 hours. FOXA2-deficient mice received intraperitoneal (i.p.) injections of saline or recombinant mouse LIF on GD 4. Scale bar: 50 μm. AM: antimesometrial, M: mesometrial, Em: embryo, LE: luminal epithelium, S: stroma. All images are representative of three independent experiments

mice (Fig. 3c). The implantation chambers on GD 5 were more ovoid in shape in saline-treated FOXA2-deficient mice and had less CLDN1 in the LE than control mice (Supplementary Figure 2b). Despite LIF repletion, the GD 5 implantation

chambers were consistently ovoid shaped in glandless *Pgr<sup>Cre/+</sup>Foxa2<sup>f/f</sup>* mice, but more spear shaped in gland-containing control and *Ltf<sup>iCre/+</sup>Foxa2<sup>f/f</sup>* mice. All examined implantation sites on GD 5 contained an embryo with a FOXA2-positive

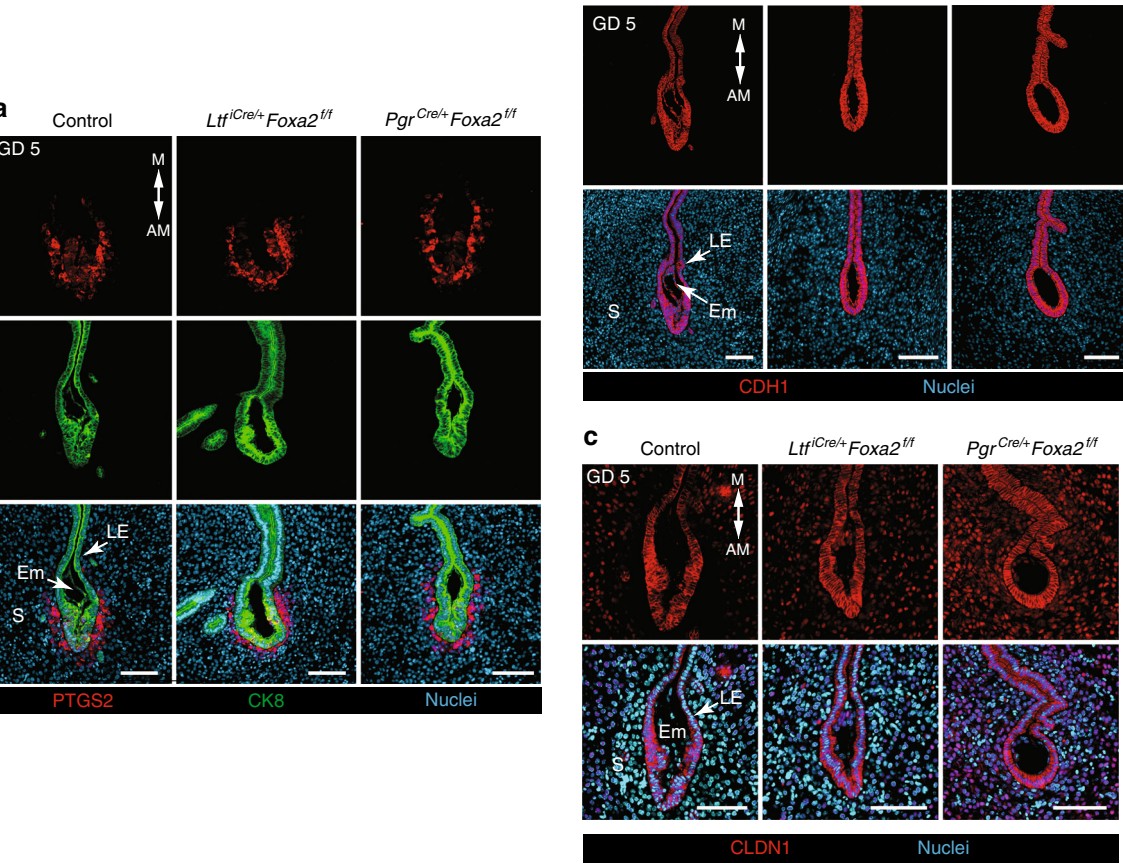

**Fig. 3** Stromal cell decidualization and epithelial remodeling in LIF-replaced FOXA2-deficient mice. Control, gland-containing $Ltf^{iCre/+}Foxa2^{f/f}$ and glandless $Pgr^{Cre/+}Foxa2^{f/f}$ mice received i.p. injections of recombinant mouse LIF on GD 4 and were evaluated on GD 5 at 0800 hours. **a** Immunofluorescence analysis of PTGS2 and CK8 in implantation sites. PTGS2 is present in the stroma around the implanting embryo on the antimesometrial side of the uterus. Scale bar: 100 μm. **b** Immunofluorescence analysis of CDH1 in implantation sites. Scale bar: 100 μm. **c** Immunofluorescence analysis of CLDN1 in implantation sites. Scale bar: 100 μm. AM: antimesometrial, M: mesometrial, Em: embryo, LE: luminal epithelium, S: stroma. All images are representative of three independent experiments

endoderm (Supplementary Figure 3a). As in humans, a recent study employing tridimensional visualization found that uterine glands remain intact during pregnancy establishment in mice[4]. Similarly, we observed that glands were present on GD 5 and appeared to connect directly into the implantation crypt (Supplementary Figure 3b).

**Implantation is disrupted in LIF-replaced glandless mice.** Active removal of LE by entosis begins on the lateral sides of the implantation chamber on GD 5 (1800–2000 hours)[7]. Removal of the LE was observed in implantation sites from both control and LIF-replaced $Ltf^{iCre/+}Foxa2^{f/f}$ mice on GD 5 at 2000 hours (Fig. 4a). The remaining LE cells surrounding the embryo displayed decreased apicobasal polarity. In contrast, intact LE cells were consistently observed in the implantation sites of LIF-replaced glandless $Pgr^{Cre/+}Foxa2^{f/f}$ mice.

Epithelial integrity was determined by evaluating adherens (CDH1) and tight junctions (CLDN1). At 2000 hours on GD 5, interrupted areas of CDH1-positive LE were observed in the lateral sides of the implantation crypts of both control and LIF-replaced $Ltf^{iCre/+}Foxa2^{f/f}$ mice, but not LIF-replaced glandless $Pgr^{Cre/+}Foxa2^{f/f}$ mice (Fig. 4b). Similarly, CLDN1-positive LE cells were absent from the lateral sides of the implantation crypts in uteri from both control and LIF-replaced $Ltf^{iCre/+}Foxa2^{f/f}$ mice

(Fig. 4c). In contrast, CLDN1 remained detectable in all LE cells on the lateral sides of the embryo in LIF-replaced glandless $Pgr^{Cre/+}Foxa2^{f/f}$ mice (Fig. 4c). However, by the morning of GD 6, the LE was completely removed from the implantation sites in LIF-replaced glandless $Pgr^{Cre/+}Foxa2^{f/f}$ mice (Fig. 5a). Collectively, these results support the idea that on-time implantation is perturbed in glandless $Pgr^{Cre/+}Foxa2^{f/f}$ mice despite LIF repletion, as evidenced by defective removal of the LE. Another possibility is that the remaining LE is a byproduct of the implantation defect and not the causation.

**Decidualization failure in mice lacking uterine glands.** By the morning of GD 6, implantation sites in both types of LIF-replaced FOXA2-deficient mice were histologically normal and virtually indistinguishable from control mice (Fig. 5a). The embryo was positioned more centrally within the uterus, LE cells were absent from the implantation chamber, and PTGS2-positive decidualizing stromal cells forming the Pdz were evident (Fig. 5b). Evidence of secondary decidual zone (Sdz) formation was found in control and FOXA2-deficient mice based on Ki67 positive proliferating cells adjacent to the Sdz in the antimesometrial regions of the implantation sites (Fig. 5c).

Although embryos were clearly present within implantation sites of $Pgr^{Cre/+}Foxa2^{f/f}$ mice on GD 7, their morphology was more

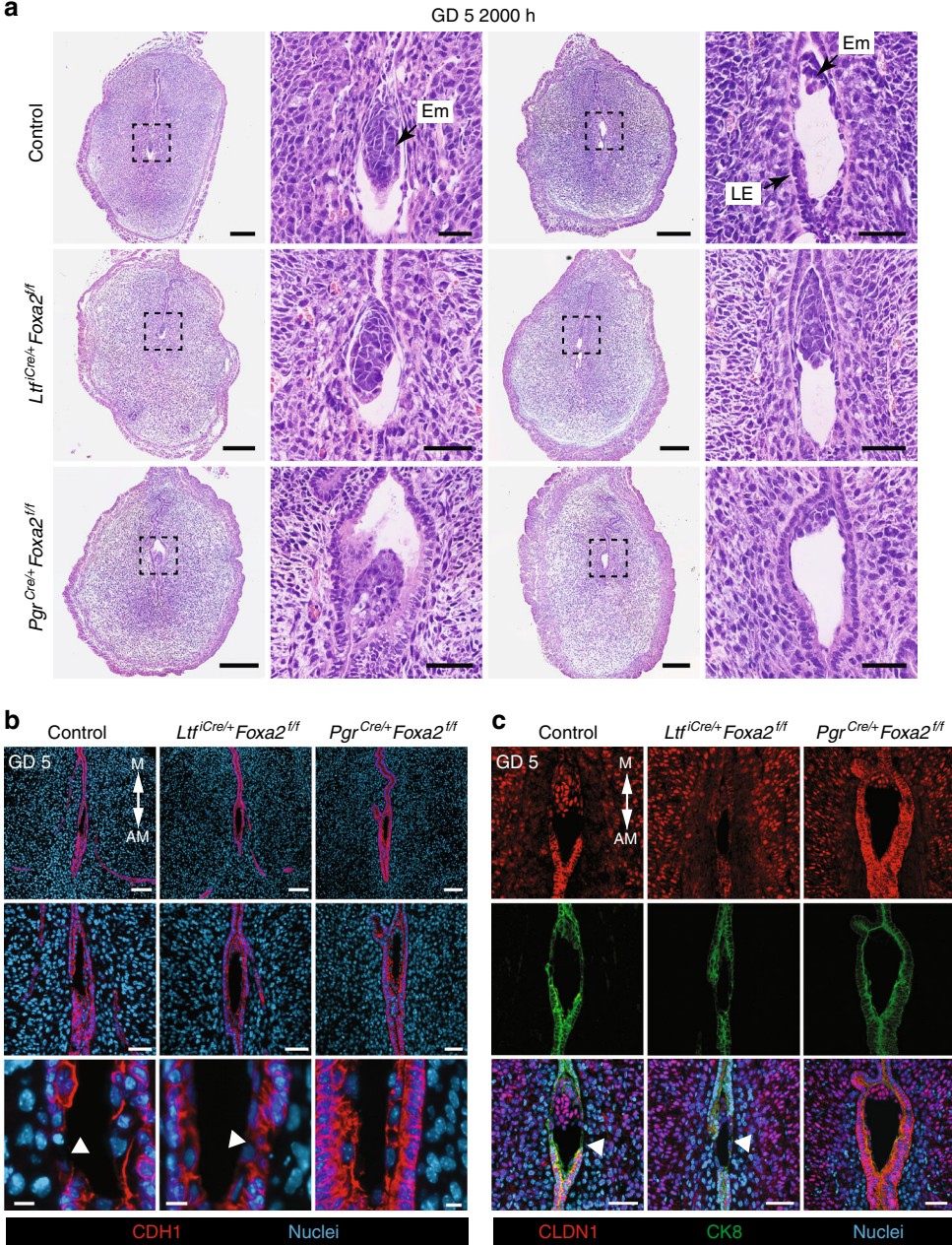

**Fig. 4** On-time removal of the LE is defective in LIF-replaced glandless $Pgr^{Cre/+}Foxa2^{f/f}$ mice. Control, gland-containing $Ltf^{iCre/+}Foxa2^{f/f}$ and glandless $Pgr^{Cre/+}Foxa2^{f/f}$ mice received i.p. injections of recombinant mouse LIF on GD 4 and were evaluated at 2000 hours on GD 5. **a** Section of implantation sites on GD 5 were stained with hematoxylin and eosin. Representative images are shown for two independent mice representing observed variation. Columns 1 and 3—Scale bar: 250 µm; Columns 2 and 4—Scale bar, 50 µm. **b** Immunofluorescence analysis of CDH1 in implantation sites. Areas of breached epithelium are indicted with white arrowheads. Row 1—Scale bar: 100 µm; Row 2—Scale bar: 50 µm; Row 3—Scale bar: 10 µm. **c** Immunofluorescence analysis of CLDN1 in implantation sites. Areas of breached epithelium are indicted with white arrowheads. Scale bar: 50 µm. AM: antimesometrial, M: mesometrial, Em: embryo, LE: luminal epithelium. All images are representative of three independent experiments

variable and less developed than in uteri of control and $Ltf^{iCre/+}Foxa2^{f/f}$ mice (Fig. 5a). PTGS2 expression was also more variable and less consistent in LIF-replaced glandless $Pgr^{Cre/+}Foxa2^{f/f}$ mice on GD 7 (Fig. 5c). Cell proliferation declined in the Sdz of implantation sites from both control and LIF-replaced $Ltf^{iCre/+}Foxa2^{f/f}$ mice; however, Ki67-positive proliferating cells remained in the Pdz and Sdz of implantation sites from LIF-replaced glandless $Pgr^{Cre/+}Foxa2^{f/f}$ mice on GD 7 (Fig. 5c). Delayed embryo development and embryo resorption was particularly noticeable by GD 8 in LIF-replaced $Pgr^{Cre/+}Foxa2^{f/f}$ mice (Fig. 5a). Note the

substantial decreased numbers of implantation sites in uteri from LIF-replaced $Pgr^{Cre/+}Foxa2^{f/f}$ mice (Fig. 5d). TUNEL analysis revealed that implantation sites of LIF-replaced $Pgr^{Cre/+}Foxa2^{f/f}$ mice contained considerable amounts of apoptotic decidual cells as well as degenerating embryos (Fig. 5d). Complete resorption of embryos and implantation sites are observed by GD 10 in LIF-replaced glandless $Pgr^{Cre/+}Foxa2^{f/f}$ mice[16]. These results support the idea that adverse ripple effects of asynchronous implantation and defective decidualization results in embryo resorption and pregnancy loss in glandless $Pgr^{Cre/+}Foxa2^{f/f}$ mice.

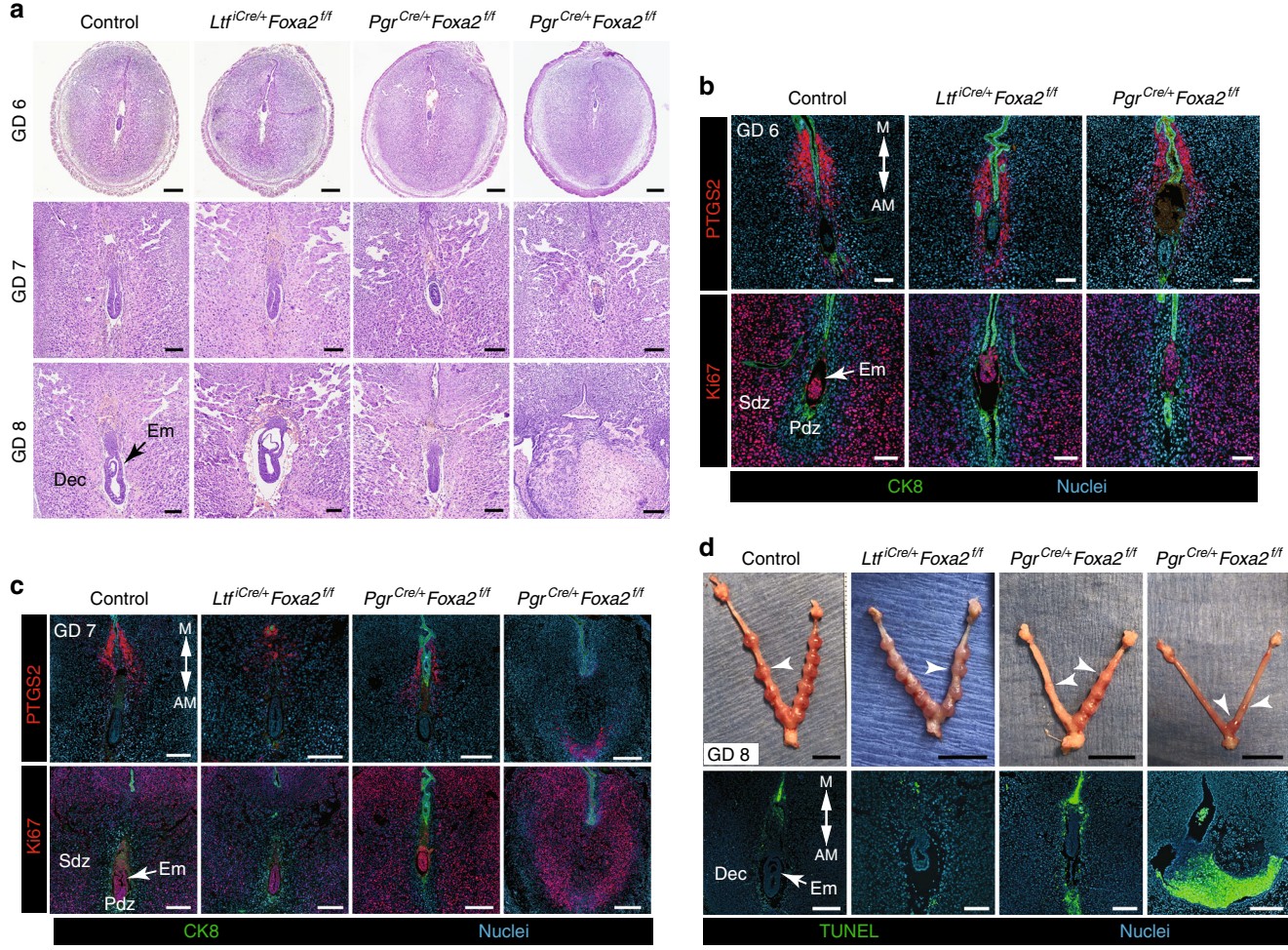

**Fig. 5** Uterine glands are essential for decidual progression and pregnancy establishment in mice. Control, gland-containing *Ltf^iCre/+Foxa2^f/f* and glandless *Pgr^Cre/+Foxa2^f/f* mice received i.p. injections of recombinant mouse LIF on GD 4 and were evaluated on GDs 6, 7, and 8. **a** Sections of implantation sites were stained with hematoxylin and eosin. Note the variation in embryo and decidua development on GDs 7 and 8 in the glandless *Pgr^Cre/+Foxa2^f/f* uterus. Row 1—Scale bar: 250 μm; Row 2—Scale bar: 150 μm; Row 3—Scale bar: 150 μm. **b** Immunofluorescence analysis of PTGS2 and Ki67 (cell proliferation marker) in implantation sites on GD 6. Scale bar: 100 μm. **c** Immunofluorescence analysis of PTGS2 and Ki67 (cell proliferation marker) in implantation sites on GD 7. Scale bar: 250 μm. **d** Gross morphology of the uterus on GD 8 (top panel). White arrowheads indicate individual implantation sites. Scale bar: 1 cm. TUNEL staining to detect apoptosis in implantation sites on GD 8 (bottom panel). Scale bar: 250 μm. Em: embryo, Dec: decidua, Pdz: primary decidual zone, Sdz: secondary decidual zone. All images are representative of three independent experiments

**Glands influence the GD 6 uterine transcriptome**. Transcriptome profiling of implantation sites on GD 6 was conducted and found that expression of 1332 genes differed (552 increased, 780 decreased) between control and LIF-replaced *Pgr^Cre/+Foxa2^f/f* mice on GD 6 (Supplementary Data 5). Many of the 780 decreased genes are known to be expressed uniquely in uterine glands (*Cxcl15, Foxa2, Prss28, Prss29, Sox9, Spink3, Ttr*) and thus are not expressed in the uterus of LIF-replaced glandless *Pgr^Cre/+ Foxa2^f/f* mice (Fig. 6a). Further, 53 of the differentially expressed genes are implicated or known to be involved in stromal cell decidualization (*Bmp7, Cdh1, Cebpb, Ptgs2, Ptx3, Wnt4*)[31–37] (Fig. 6b). Functional analysis found that the differentially expressed genes were enriched in a number of biological processes including cell adhesion, cell differentiation, cellular proliferation, immune response, and vasculature development (Fig. 6c and Supplementary Data 6).

Next, genes were identified that increased or decreased in the uterus of control and glandless LIF-replaced *Pgr^Cre/+Foxa2^f/f* mice from GD 4 to GD 6 (Supplementary Figure 4 and Supplementary Data 7 and 8). Functional analysis of the 4616

genes that increased in control mice were enriched for biological processes that included cell cycle and reproduction and pathways associated with cell cycle, extracellular matrix, steroid and biosynthesis among others (Supplementary Data 9). Many of those increased genes are known to be expressed in decidualized stromal cells, such as members of the prolactin (PRL) family (*Prl8a2, Prl3c1, Prl8a9, Prl3d1, Prl3d2, Prl6a1, Prl3d3, Prl7b1, Prl5a1*), enzymes (*Ass1, Ptgs2, Sulf1, Fads3*) and secreted ligands (*Bmp2, Wnt4*). Further, several are expressed by uterine glands (*Hp, Prss28, Prss29, Spink1, Tdo2*). Functional analysis of the 3289 decreased genes that decreased in control mice found enrichment in pathways such as extracellular matrix and cell adhesion molecules (Supplementary Data 10). Several of the decreased genes are also expressed in the GE (*Cxcl15, Foxa2, Lif, Sult1d1, Ttr*) and LE (*Cldn1, Krt8, Ltf, Wnt7a, Wnt11*).

In the uteri of LIF-replaced glandless *Pgr^Cre/+Foxa2^f/f* mice, 4503 genes increased and 4502 genes decreased between GDs 4 and 6. Functional analysis of those genes identified many of the same biological processes and pathways as in control uteri such as cell cycle (Supplementary Data 11 and 12). A subset of 112 genes were

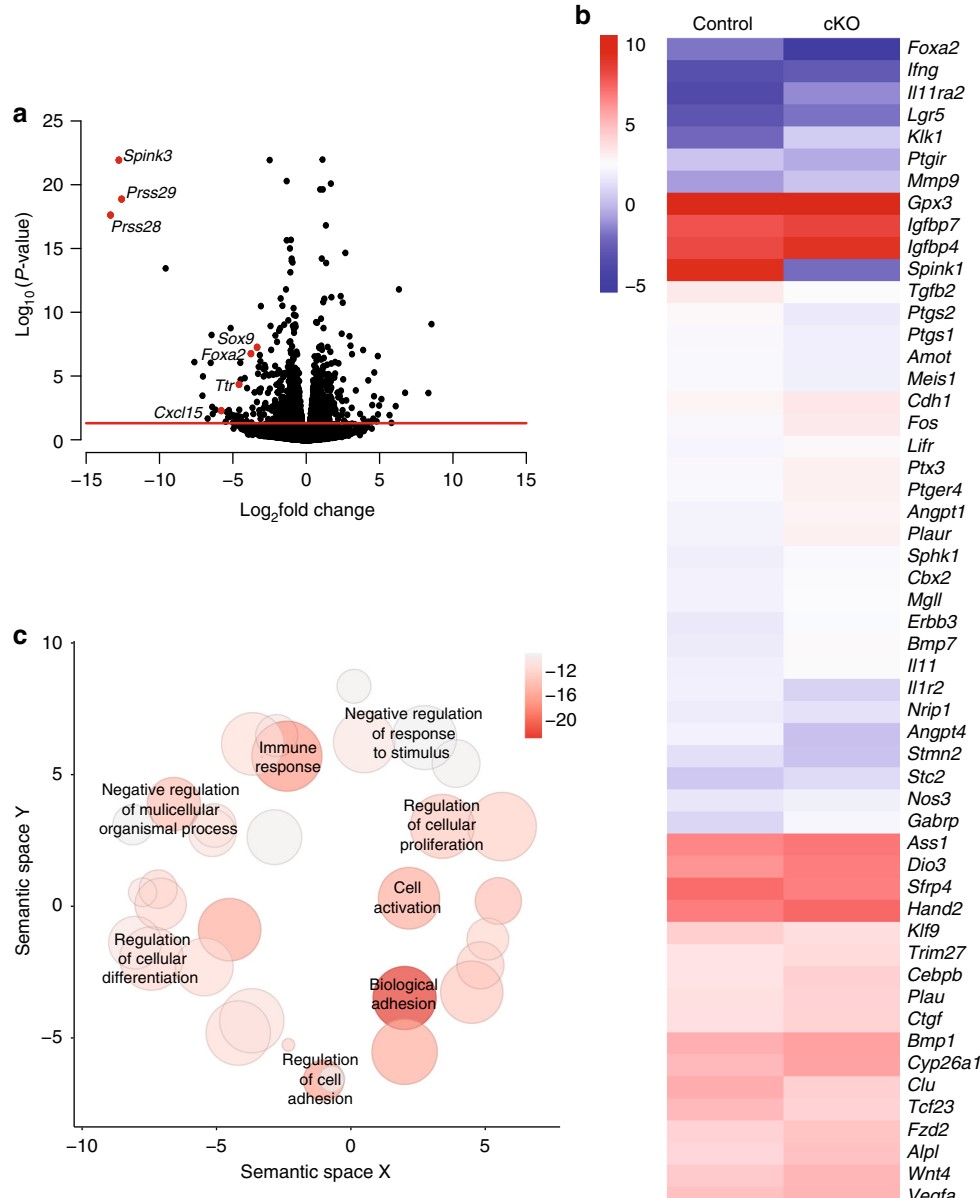

**Fig. 6** The uterine transcriptome is dysregulated in the implantation sites of GD 6 mice that lack endometrial glands. RNA-sequencing analysis was conducted using implantation sites from LIF-replaced glandless and control mice on GD 6 ($n = 4$ per genotype). **a** Volcano plot of all genes detected in transcriptome analysis. All data points above the red horizontal line are significant. Known gland-specific genes have red points and are labeled with gene names. **b** Heatmap for 53 genes (log2 FPKM values) known to be involved in decidualization that were different in GD 6 implantation sites in LIF-replaced glandless $Pgr^{Cre/+}Foxa2^{f/f}$ mice. **c** Visualization of biological process GO terms associated with genes differentially expressed in LIF-replaced glandless $Pgr^{Cre/+}Foxa2^{f/f}$ and control mice on GD 6. Color indicates the FDR $P$ value and size indicates the frequency of the GO term in the underlying annotation database. Darker circles with a larger size represent highly significant terms that are more general

found to decrease in control mice but increase in glandless LIF-replaced $Pgr^{Cre/+}Foxa2^{f/f}$ mice between GDs 4 and 6 (Supplementary Figure 4 and Supplementary Data 13). Functional analysis revealed that those 112 genes are enriched for ECM components (Supplementary Data 14). These transcriptome data provide an important resource for studies of implantation and pregnancy establishment. Collectively, results support the hypothesis that uterine glands secrete molecules that modulate on-time embryo implantation and impact stromal cell decidualization.

## Discussion

Blastocysts enter the uterus early on GD 4, and the attachment reaction is initiated within implantation crypts surrounded by glands in the antimesometrial region of the uterus around mid-night on GD 4[1,3,4,19,23]. Implantation can occur in mice that exhibit aberrant crypt formation, but this results in increased embryo resorption and pregnancy loss[1,19,20]. The present studies did not find any evidence of altered crypt formation or embryo positioning in gland-containing or glandless FOXA2-deficient mice with or without LIF replacement. Thus, uterine glands and, specifically, LIF is not involved in implantation crypt formation or blastocyst positioning within the uterus.

Failure of embryo attachment in glandless $Pgr^{Cre/+}Foxa2^{f/f}$ and gland-containing $Ltf^{iCre/+} Foxa2^{f/f}$ supports the idea that embryo−uterine interactions are perturbed in the absence of LIF. $Hbegf$, one of the first established molecular mediators of blastocyst-

uterine interactions[25], was not upregulated in the LE of the implantation crypt in FOXA2-deficient mice. However, LIF repletion was sufficient to induce expression of *Hbegf* and activated STAT3 in the LE of the implantation crypt, marking the onset of blastocyst attachment to the LE and communication with the uterus. *Lif receptor* (*Lifr*) null embryos implant and establish pregnancy that continues to term[38], but mice with a conditional specific deletion of uterine either *Lifr* or *Stat3* in the epithelia are infertile due to defects in embryo attachment[28,39]. Collective available evidence supports the idea that uterine glands secrete LIF that acts on the LE to induce HBEGF in response to the embryo and enable firm attachment and adhesion of the activated blastocyst trophectoderm to the uterine LE for implantation in mice. The present studies support the idea that uterine glands do not directly influence blastocyst activation for attachment to the receptive LE for nidation in mice, as auto induction of HBEGF requires implantation competent blastocysts[40]. In humans, there is limited information of LIF in the endometrium, but it is expressed in the glands of secretory phase endometrium[41] and implicated in embryo implantation based on in vitro studies[42]. Further, less LIF is present in uterine luminal fluid from infertile as compared to normal fertile women[43].

Following initiation of the attachment reaction, embryo implantation proceeds on the morning of GD 5 as stromal cells adjacent to the LE of the implantation crypt undergo proliferation and differentiation into epithelioid like decidual cells[44]. Decidualization is a critical pregnancy event that regulates placental development and supports embryonic growth[5]. PTGS2 is a marker of stromal cell decidualization and important for formation of the Pdz[24,45]. Of note, the present study found that PTGS2 was upregulated in the LE adjacent to the unattached embryo on GD 5 in FOXA2-deficient mice that lack LIF, yet HBEGF was not upregulated in the LE unless LIF was repleted on GD 4. Prostaglandins are important for implantation, and *Ptgs2* is expressed in both the LE and stroma surrounding the implanting blastocyst in mice[24]. In mouse mutants with reduced fertility, *Lif* null and *Msx1/Msx2* double conditional knockouts, PTGS2 expression is also observed only in the LE, suggesting that proper communication between the LE and stroma is essential for embryo nidation and pregnancy establishment[19,46]. Thus, the peri-attachment blastocysts must produce a paracrine-acting factor(s) that stimulates PTGS2 expression before actively removing the LE by entosis.

In the present study, LIF repletion corrected the defect in PTGS2 expression in the Sdz of uteri in both glandless and gland-containing FOXA2-deficient mice. Subsequently, pregnancy loss was observed only in LIF-replaced glandless mice, suggesting functional coordination between the blastocyst, epithelium and stroma requires LIF and other undetermined secretions of the glands. Of note, the implantation crypts were mostly ovoid in shape on the morning of GD 5 in LIF-replaced glandless mice as compared to spear shaped in control and LIF-replaced gland-containing mice. The ovoid implantation crypt shape is predictive of pregnancy loss[4] and is an indicator of defective on-time implantation with subsequent pregnancy failure in the LIF-replaced glandless FOXA2-deficient mice.

The LE cells in direct contact with the blastocyst are actively removed beginning on GD 5 between 1800 and 2000 hours by the trophoblast cells using a nonapoptotic cell-in-cell invasion process termed entosis[7]. Removal of the LE adjacent to the blastocyst allows for trophectoderm cells to make direct physical contact with the decidualizing stroma during nidation. Here, the LE remained intact on GD 5 at 2000 hours in LIF-replaced glandless *Pgr^{Cre/+}Foxa2^{f/f}* but not in control or LIF-replaced gland-containing *Ltf^{iCre/+}Foxa2^{f/f}* mice. This result strongly supports the posit that uterine glands secrete factors other than LIF that

accelerate entosis for on-time implantation. Alternatively, the remaining intact LE in on GD 5 in LIF-replaced glandless mice could by a byproduct of the implantation defect rather than the causation.

Our transcriptomics analysis here is a first step in identifying the unknown gland-derived factor(s) influencing other uterine cell types and perhaps the embryo during pregnancy establishment. Analysis of ligands enriched in the GE of control mice, and exclusively decreased in the uteri of *Pgr^{Cre/+}Foxa2^{f/f}*, identified fibronectin (*Fn1*), whose cognate receptor is also downregulated in the uterus of glandless mice. Fibronectin is implicated in fertility as it is a ligand for integrin αvβ3 that is proposed to mediate embryo attachment in mice and other mammals (human, rabbit, and domestic animals)[47,48]. Additionally, integrin α5β1-fibronectin engagement induces calmodulin-mediated calcium transients in the blastocyst and fibronectin induces trafficking of αIIbβ3, which is likely involved in trophoblast invasion[49,50]. Fertility has not been assessed in *Fn1* mutant mice as null mutants are embryonic lethal[51]. Conditional knockout of Fn1 and other identified FOXA2-independent factors in the uterine glands would begin to define molecular crosstalk between uterine glands, stroma, and blastocysts that occur during uterine receptivity and embryo implantation and important for pregnancy establishment.

On GD 6, implantation sites appeared normal in LIF-replaced glandless mice (*Pgr^{Cre/+}Foxa2^{f/f}*) based on morphology (decidual swellings), histology, cell proliferation (Ki67) and molecular markers (PTGS2). However by GD 7, defects in embryo development and decidual regression were clearly evident in LIF-replaced glandless mice with full embryo loss and resorption by GD 10[16]. In contrast, pregnancies are maintained to term in gland-containing *Ltf^{iCre/+}Foxa2^{f/f}* mice receiving LIF repletion on GD 4[16]. Transcriptome analysis of GD 6 implantation sites revealed numerous genes and pathways altered in the uterus of control as compared to LIF-replaced glandless mice. Many of those genes have known or purported biological roles in uterine stromal cell decidualization. For instance, *Bmp7*, *Cdh1*, *Cebpb*, *Ptgs2*, *Ptx3*, and *Wnt4* conditional knockout mouse models display decidualization defects in their uterus similar to findings in LIF-replaced glandless mice[10,16,31–37,52]. The majority of those known decidualization-related genes are increased in implantation sites of glandless uteri (*Bmp7*, *Cdh1*, *Cebpb*, *Ptx3*, *Wnt4*), which is indicative of premature stromal cell differentiation into decidual cells. Decreased expression of *Bmp7* occurs as decidualization and placentation progresses in both mice and humans[31,53]. The increased expression of genes crucial to decidualization on GD 6 and persistence of stromal proliferation on GD 7 suggests perturbed decidual progression and premature decidual senescence in mice lacking uterine glands. It is tempting to speculate that uterine glands produce paracrine-acting factors that govern stromal cell decidualization by modulating proliferation, differentiation and(or) polyploidization[6]. The concept of gland-derived factors influencing decidualization at the implantation site in mice and humans is novel and underexplored, but supported by evidence that LIF enhances in vitro endometrial stromal cell decidualization in both mice and humans[54].

Our transcriptomics analyses of the uterus continually identified alterations in pathways related to ECM remodeling, an essential occurrence during decidualization[55], suggesting that gland-derived factors impact structural remodeling of the implantation site in response to pregnancy. In both mice and humans, uterine glands are present throughout pregnancy in the endometrium surrounding the implantation sites and within the developing decidua[4,8,56], but little is known about what they express and secrete in either species. Our present and previous findings demonstrate that LIF replacement alone is sufficient for

pregnancy establishment and maintenance in gland-containing FOXA2-deficient mice[16]. Future studies should focus on discovering and understanding the function of factors produced by uterine glands during embryo nidation and decidualization in mice and humans[4]. The studies here posit that those factors are critical for on-time implantation and decidualization, which are primary determinants of pregnancy establishment and success[3,55,57,58].

For many decades, secretions of glands have been hypothesized to play an essential role in sustaining the conceptus before implantation in several species[59], but their role in post-implantation growth and development of the embryo and placenta was largely ignored[4,8,60]. Mounting evidence from studies in mice and humans over the past decade supports the idea that uterine glands and their secretions have fundamental biological roles in uterine receptivity, blastocyst implantation, and post-implantation fetal and placental development[9,60,61]. Often unrecognized, the implantation site is replete with functional glands during the first trimester of human pregnancy[56], and those glands have the potential to interact with multiple cell types, including the decidua, vasculature, immune cells, and trophoblast[62]. Given that uterine gland dysfunction may cause pregnancy loss and complications such as miscarriage, preeclampsia, and fetal growth retardation[8,63], increased knowledge of uterine glands may provide diagnostic and prognostic markers of endometrial function and pregnancy complications useful in natural and assisted reproduction[9].

## Methods

**Animals**. All animal procedures were approved by the Institutional Animal Care and Use Committee of the University of Missouri and were conducted according to NIH Guide for the Care and Use of Laboratory Animals. Floxed *Foxa2* (*Foxa2*$^{f/f}$) mice[64] were crossed with *Pgr*$^{Cre}$[65] or *Ltf*$^{iCre}$[66] mice to generate conditional knockout animals. *Foxa2*$^{f/f}$ mice (*Foxa2*$^{tm1Khk}$) were obtained from The Jackson Laboratory (stock no. 022620). *Ltf*$^{iCre}$ mice (*Ltf*$^{tm1(icre)Tdku}$) were obtained from the Jackson Laboratory (stock no. 026030). *Pgr*$^{Cre}$ mice (*Pgr*$^{tm2(cre)Lyd}$) were generously provided by John Lydon (Baylor College of Medicine, Houston, Texas). Gestational time points were obtained by the mating of randomly selected 8- to 10-week-old females of *Foxa2*$^{f/f}$ control or *Foxa2* conditional knockout animals with CD-1 male mice with the day of vaginal plug observation considered GD 1. For rescue of implantation, FOXA2-deficient mice received i.p. injections of recombinant mouse LIF (10 μg in saline; catalog #554008, BioLegend) and control mice received saline on GD 4 at 1000 and 1800 hours. Control and LIF-replaced mice were then housed separately until implantation analysis. Implantation sites on the morning of GD 5 were visualized by intravenous injection of 1% Evans blue dye (Sigma-Aldrich) before necropsy. At necropsy, uteri were excised, trimmed of fat, washed with PBS, and then frozen in liquid nitrogen or fixed with 4% paraformaldehyde. To confirm pregnancy in plug-positive females with no visual implantation sites, one uterine horn was flushed for the presence of blastocysts. Based on the consistency of results and our previous publications, each experiment was repeated five times.

**Histology and immunolocalization**. At least three implantation sites were examined per mouse (*n* = 3). As previously described[16], fixed uteri were sectioned (5 μm), mounted on slides, deparaffinized in xylene, and rehydrated in a graded alcohol series. Antigen retrieval was performed by incubating sections in boiling 10 mM citrate buffer (pH 6.0) for 10 min. Sections were then blocked with 5% (vol/vol) normal goat serum (Catalog # 50062Z, Thermo Fisher Scientific) in PBS (pH 7.2) at room temperature for 1 h and then incubated with primary antibodies (see Supplementary Table 2) overnight at 4 °C in 1% BSA in PBS. Sections were washed in PBST (PBS with 0.05% Tween 20) and incubated with 5 μg/ml biotinylated secondary antibody raised in goat (Catalog # PK-6101, Vector Laboratories) for 1 h at 37 °C in PBS. Immunoreactive PTGS2 (1:250, 160106, Cayman Chemicals) was visualized using a Vectastain ABC kit (Catalog # PK-6101, Vector Laboratories) and diaminobenzidine tetrahydrochloride as the chromagen. Sections were lightly counterstained with hematoxylin before affixing coverslips with Permount. Immunofluorescence for CDH1 (1:400, ECCD-2, Thermo Fisher Scientific), CK8 (1:50, University of Iowa Developmental Studies Hybridoma Bank), CLDN1 (1:600, MH25, Thermo Fisher Scientific), FOXA2 (1:1000, ab108422, Abcam), Ki67 (1:1000, ab15589, Abcam), p-Tyr705 STAT3 (1:300, BS4181, Bioworld Technology), PTGS2 (1:250, 160106, Cayman Chemicals) was performed using Alexa 488 or Alexa 555-conjugated secondary antibodies (1:400 dilution; Catalog #A-11034 or A-31572, Thermo Fisher Scientific) incubated for 90 min at room temperature. Sections were counterstained with Hoechst 33342 (2 μg/ml;

Catalog # H3570, Life Technologies). Brightfield and fluorescent images were collected with a Leica DM5500 B upright microscope and Leica DFC450 C camera using Leica Application Suite X (LAS X).

**Detection of DNA damage by TUNEL assay**. Apoptotic cells were detected in sections of mouse uteri using the DeadEnd Fluorometric TUNEL kit (Catalog # G3250, Promega).

**In situ localization of *Hbegf* mRNA**. RNAscope in situ hybridization (Advanced Cell Diagnostics) was performed according to the manufacturer's instructions using paraformaldehyde-fixed tissues and a mouse *Hbegf* probe (Catalog # 437601). Following hybridization, slides were washed and probe binding visualized using the HD 2.5 Red Detection Kit (Catalog # 322350, Advanced Cell Diagnostics). Sections were briefly counterstained with hematoxylin before dehydrating and affixing coverslips with Permount.

**RNA extraction and transcriptome analysis**. Total RNA was isolated from frozen uteri using a standard TRIzol-based protocol (Catalog 15596026, Thermo Fisher). To eliminate genomic DNA contamination, extracted RNA was treated with DNase I and purified using an RNeasy MinElute Cleanup Kit (Qiagen). Quality and concentration of RNA were determined using a Fragment Analyzer (Advanced Analytical Technologies). Libraries were prepared by the University of Missouri DNA Core Facility using an Illumina TruSeq mRNA kit (Illumina Inc.) and sequenced (2 × 75 base pair paired end) using an Illumina NextSeq 500. Adapters were trimmed from reads using cutadapt (version 1.11) and quality trimmed to a sliding window quality score of 30 and minimum length of 20 bp with fqtrim software. Reads were mapped to the *Mus musculus* genome assembly (GRCm38.p5) using HISAT2 (version 2.0.3)[67]. Reads overlapping Ensembl annotations were quantified with featureCounts (version 1.5.0)[68]. Genes with evidence of expression (counts per million; CPM rowSum > 0) were used for model-based differential expression (DE) analysis using the edgeR-robust method[69]. Differentially expressed gene list (FDR < 0.05) enrichment analysis was conducted using ToppFun (https://toppgene.cchmc.org/) with default settings[70]. The Bonferroni procedure was used to control FDR for GO term and pathway enrichment analyses. Raw FASTQ files were deposited in the NCBI Gene Expression Omnibus (GSE113065).

**Ligand-receptor analysis**. Differentially expressed genes unique to the uteri of *Pgr*$^{Cre/+}$*Foxa2*$^{f/f}$ and enriched in the GE[21] identify ligands and receptors from the curated ligand−receptor information within the FANTOM5 database[22].

**Data availability**. The data that support the findings of this study are available from the article and Supplementary Information files, or from the corresponding author upon request. Raw FASQ data files are publically available in the NCBI Gene Expression Omnibus (GSE113065).

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

## Acknowledgements

This work was supported in part by Grant R21 HD076347 and R01 HD096266 from the Eunice Kennedy Shriver National Institute of Child Health and Development.

## Author contributions

A.M.K. and T.E.S. conceived the study and designed experiments; A.M.K. and J.M.-F. performed the experiments; A.M.K. performed histology imaging; S.K.B. and A.M.K. performed transcriptomic data analysis; A.M.K. and T.E.S. wrote the paper; and T.E.S. supervised the entire project.

## Additional information

**Competing interests:** The authors declare no competing interests.

