## [Peer Review File · Nature Communications]

Reviewers' comments:

Reviewer #1 (Remarks to the Author):

The study by Kelleher et al., presents a detailed analysis of two mouse models with conditional deletion of FOXA2. In this study, the role of FOXA2 was determined based on its conditional deletion from 1) all uterine compartments (with Pgr-cre) or 2) from cells of the uterine epithelium (with Ltf-cre). The studies in this manuscript present a significant advance in our understanding of the roles of uterine glands in post-implantation biology. Most significantly, these studies demonstrate that although the deleterious effects of uterine glands on implantation can be rescued with LIF supplementation, pregnancy cannot be carried to term in the absence of uterine glands. The authors use detailed histological, molecular, and transcriptomic analyses to reach this conclusion. The premise for each experiment is well supported by the literature, and the findings are placed into larger context in the discussion. Furthermore, they identify novel glandular factors that may be of significance to implantation and post-implantation processes in mice and humans.

Minor comments about the manuscript are below,

- 1) Line 200-202: the authors state that the implantation defects were due to improper removal of the luminal epithelium (LE) adjacent to the implanting embryo. It is possible that the remaining LE is a byproduct of the implantation defect and not the causation? This possibility should be stated in the results and further addressed in the discussion.
- 2) The introduction and results mention GD but never mention when GD0 starts (is it at mating and fertilization) and describes "midnight" or afternoon (line 87). These should be explained in more detail.
- 3) In the introduction, the authors mention PR-Cre and LTF-Cre, but these should be explained in more detail (cell types affected and temporal deletion).
- 4) The authors should mention anything that is known about the role of LIF in humans.
- 5) The introduction is a little awkward and is not written for a general audience.
- 6) The terminology used to refer to the luminal epithelial breakdown at the site of blastocyst attachment could be revised to more accurately describe the biological process. In the abstract and throughout the manuscript, the authors refer to luminal epithelial breakdown as "removal of the luminal epithelium," which implies a passive process. In fact, as the authors state, this is a highly active biological process that involves entosis of the luminal epithelium by the invading trophoctoderm. The terminology should be revised throughout to reflect this active process.
- 7) One novel aspect of this study is the inability of the glandless, LIF-supplemented uterus to carry a pregnancy beyond 10 days of gestation. This finding is interesting given that it has implications for problems that may arise in mid- to late-gestation in women, such pre-

eclampsia. It also implies that unidentified glandular-specific secretions are required to support embryonic development. Because this mouse model can be used to identify this "unknown" factor, further characterization of the GD 6-10 implantation sites is important to unravel how pregnancy is affected in the glandless mice. The authors used TUNEL assays and Ki67 staining in GD6-8 implantation sites, but other markers of implantation site development should be analyzed. Implantation site markers such as, alkaline phosphatase (decidualization), vascularization (SMA, VEGF, ANGPT, laminin), immune cell infiltration (uNKs, DBA, PAS) or extraembryonic structure development (MMP9, PRL3D1). These markers would provide valuable information about the developmental processes affected by the absence of glands at mid-gestation.

8) The authors mention that the ovoid shape of the implantation sites in FOXA2 glandless mice is predictive of pregnancy loss (line 324). Estrogen and progesterone are critical during this time point and for creating a receptive endometrium. Did the authors check estrogen receptor and progesterone receptor expression and their downstream targets? It is important to know whether LIF supplementation is not sufficient for restoring the response to estrogen and progesterone, especially in the glandless mouse model.

Reviewer #2 (Remarks to the Author):

SUMMARY: Using FOXA2-deficient conditional knockout mouse models, genome-wide transcriptomic profiling, and in-depth immunohistochemical and molecular analysis, Kelleher et al. provide compelling support for a critical role for glandular epithelia of the endometrium in embryo implantation and subsequent endometrial decidualization, essential uterine events in the early establishment of pregnancy in mammals.

CRITIQUE: This is an extremely well-written and organized manuscript that describes an important set of experiments that convincingly defines the involvement of the glandular epithelial compartment in early pregnancy establishment, an understudied area of female reproductive biology that may provide new insights into infertility caused by a dysfunctional uterus. That said, a number of issues need to be addressed.

Major: In Fig. 1a, it is difficult to see the hbegf in situ hybridization positive signal in control panel (GD4). A higher magnification and/or indicate by arrow the location of the signal. The transcriptomic studies in Fig. 2 are described before completion of the discussion of the results displayed in Fig 1b and 1c. This format breaks up the logical flow of the results section, needs to be revised.

Minor: The Discussion should be shortened (nearly five pages in its present form). The initial section of the Discussion, which repeats parts of the Results section, could be removed. Also, the subsection titles within the results section could be more informative to indicate the ensuing results being described.

Reviewer #3 (Remarks to the Author):

A previous publication using glandless *Pgr cre/+ Foxa2 f/f* mice and gland-containing *Ltf-cre Foxa2 f/f* mice revealed critical role of *Foxa2* in pregnancy establishment. Based on previous study, the current study by Kelleher AM et al. confirmed that removal of the luminal epithelium is delayed and subsequent decidualization fails in LIF-replaced glandless but not gland-containing *FOXA2*-deficient mice. Importantly, this study newly found that adverse ripple effects of those dysregulated events in the endometrium lacking glands resulted in embryo resorption and pregnancy failure. The study is carefully designed and meticulously executed. The results of the manuscript would help understand the function of uterine glands in embryo implantation and decidualization. Although the current study includes some novel findings, the transcriptome analysis is confused and not well presented. There are several points to be addressed.

1. Abstract: Line 35-36: The authors do not provide any evidence of uterine gland dysfunction in women.
2. Although the IHC analysis was examined in the luminal epithelium of the implantation chamber (crypt), the expression patterns of these proteins in Fig. 1b, Fig. 3a, and Suppl. Fig. 2a have been published by the previous PNAS paper.
3. The mRNA-seq results are confused and descriptive. The analysis results do not provide clear biological meaning in the implantation process.
4. Fig. 2a and b: Why are the top 150 genes selected for the heatmap analysis? Are there any specific patterns of gene expression profiling in the top genes?
5. How are "LE vs GE" genes selected from the previous data? Provide detailed method how two data sets are compared.
6. The Venn diagram are confused. Where are 2303 genes and 361 genes in the Venn diagram (Line 109-111)?
7. Suppl. Fig. 5: It is difficult to find the 2794 increased genes and the 2238 decreased genes the Venn diagram.
8. Line 201-201: The authors mentions that glandless *PgrCre/+Foxa2f/f* mice have a defect of on-time implantation due to a delay in LE removal. When do *PgrCre/+Foxa2f/f* mice show LE removal during the pregnancy?
9. The color scales of heatmap are opposite in Fig. 2 and Fig. 6. Keep the same pattern.

Point-by-Point Response to Referees' Comments

The authors appreciate the time and effort of the Referees and Senior Editor for their evaluation of our manuscript, which was found to be of interest. The verbatim comments of the Editor and referees are provided in *italics*, and our response and revisions to the manuscript provided for each comment in a point-by-point manner.

Reviewer #1 (Remarks to the Author):

The study by Kelleher et al., presents a detailed analysis of two mouse models with conditional deletion of FOXA2. In this study, the role of FOXA2 was determined based on its conditional deletion from 1) all uterine compartments (with Pgr-cre) or 2) from cells of the uterine epithelium (with Ltf-cre). The studies in this manuscript present a significant advance in our understanding of the roles of uterine glands in post-implantation biology. Most significantly, these studies demonstrate that although the deleterious effects of uterine glands on implantation can be rescued with LIF supplementation, pregnancy cannot be carried to term in the absence of uterine glands. The authors use detailed histological, molecular, and transcriptomic analyses to reach this conclusion. The premise for each experiment is well supported by the literature, and the findings are placed into larger context in the discussion. Furthermore, they identify novel glandular factors that may be of significance to implantation and post-implantation processes in mice and humans.

Author Response: Thanks!

Minor comments about the manuscript are below,

1) Line 200-202: the authors state that the implantation defects were due to improper removal of the luminal epithelium (LE) adjacent to the implanting embryo. It is possible that the remaining LE is a byproduct of the implantation defect and not the causation? This possibility should be stated in the results and further addressed in the discussion.

Author Response: This interesting possibility is stated in the Results (Lines 200-201) and further addressed in the Discussion (Lines 328-330).

2) *The introduction and results mention GD but never mention when GDO starts (is it at mating and fertilization) and describes "midnight" or afternoon (line 87). These should be explained in more detail.*

Author Response: GD 1 is the observation of a post-coital vaginal plug as defined in Line 38.

3) *In the introduction, the authors mention PR-Cre and LTF-Cre, but these should be explained in more detail (cell types affected and temporal deletion).*

Author Response: The Introduction was revised to provide more detail on the Cre lines used in the manuscript (Lines 57-62)

4) *The authors should mention anything that is known about the role of LIF in humans.*

Author Response: This information is provided in the revised Discussion (Lines 292-295 and 365-368).

5) *The introduction is a little awkward and is not written for a general audience.*

Author Response: The Introduction was reorganized and revised to address this comment.

6) *The terminology used to refer to the luminal epithelial breakdown at the site of blastocyst attachment could be revised to more accurately describe the biological process. In the abstract and throughout the manuscript, the authors refer to luminal epithelial breakdown as "removal of the luminal epithelium," which implies a passive process. In fact, as the authors state, this is a highly active biological process that involves entosis of the luminal epithelium by the invading trophoctoderm. The terminology should be revised throughout to reflect this active process.*

Author Response: The work of S.K. Dey defined entosis as the mechanism whereby LE cells were removed during implantation (Li et al., *Cell Reports* 2015), and the authors referred to this process as "removal of the luminal epithelium". This process is referred to more active than passive in the revised manuscript.

7) *One novel aspect of this study is the inability of the glandless, LIF-supplemented uterus to carry a pregnancy beyond 10 days of gestation. This finding is interesting given that it has implications for problems that may arise in mid- to late-gestation in women, such pre-eclampsia. It also implies that unidentified glandular-specific secretions are required to support embryonic development. Because this mouse model can be used to identify this "unknown" factor, further characterization of the GD 6-10 implantation sites is important to unravel how pregnancy is affected in the glandless mice. The authors used TUNEL assays and Ki67 staining in GD6-8 implantation sites, but other markers of implantation site development should be analyzed. Implantation site markers such as, alkaline phosphatase (decidualization), vascularization (SMA, VEGF, ANGPT, laminin), immune cell infiltration (uNKs, DBA, PAS) or*

extraembryonic structure development (MMP9, PRL3D1). These markers would provide valuable information about the developmental processes affected by the absence of glands at mid-gestation.

Author Response: We have previously published alterations in decidualization marker genes on GD 6 (Kelleher et al, *PNAS* 2017). Recently, we collaborated with Adrian Erlebacher (University of California-San Francisco) to evaluate immune cells (total leukocytes, macrophages, neutrophils, T cells, NK cells) and smooth muscle actin (SMA) in LIF-replaced gland-containing $Lt^{\beta Cre/+}Foxa2^{f/f}$ and glandless $Pgr^{Cre/+}Foxa2^{f/f}$ mice on GDs 6, 7 and 8. However, he found no differences in immune cells or SMA between the two mouse models. Future studies of the mouse models will evaluate vascularization and extraembryonic structure development, but those are beyond the scope of this manuscript in the authors' opinion.

8) The authors mention that the ovoid shape of the implantation sites in FOXA2 glandless mice is predictive of pregnancy loss (line 324). Estrogen and progesterone are critical during this time point and for creating a receptive endometrium. Did the authors check estrogen receptor and progesterone receptor expression and their downstream targets? It is important to know whether LIF supplementation is not sufficient for restoring the response to estrogen and progesterone, especially in the glandless mouse model.

Author Response: Our previous publications on the glandless $Pgr^{Cre/+}Foxa2^{f/f}$ mouse and the progesterone induced uterine gland knockout (PUGKO) mouse found no differences in expression or localization of steroid hormone receptors or their downstream targets (Filant et al., *Biol Reprod* 2012, Kelleher et al., *Scientific Reports* 2016, Kelleher et al., *PNAS* 2017).

Reviewer #2 (Remarks to the Author):

SUMMARY: Using FOXA2-deficient conditional knockout mouse models, genome-wide transcriptomic profiling, and in-depth immunohistochemical and molecular analysis, Kelleher et al. provide compelling support for a critical role for glandular epithelia of the endometrium in embryo implantation and subsequent endometrial decidualization, essential uterine events in the early establishment of pregnancy in mammals.

Author Response: Thanks!

CRITIQUE: This is an extremely well-written and organized manuscript that describes an important set of experiments that convincingly defines the involvement of the glandular epithelial compartment in early pregnancy establishment, an understudied area of female reproductive biology that may provide new insights into infertility caused by a dysfunctional uterus. That said, a number of issues need to be addressed.

Major: In Fig. 1a, it is difficult to see the hbegef in situ hybridization positive signal in control panel (GD4). A higher magnification and/or indicate by arrow the location of the signal. The transcriptomic studies in Fig. 2 are described before completion of the discussion of the results

displayed in Fig 1b and 1c. This format breaks up the logical flow of the results section, needs to be revised.

Author Response: As suggested by the referee, the transcriptomic data is discussed first before the implantation crypt and LIF-repletion data in the revised manuscript. A higher magnification image of *Hbegf* mRNA localization is now provided as requested (Fig. 2a).

Minor: The Discussion should be shortened (nearly five pages in its present form). The initial section of the Discussion, which repeats parts of the Results section, could be removed. Also, the subsection titles within the results section could be more informative to indicate the ensuing results being described.

Author Response: The Discussion was shortened in the revised manuscript. Further, the subsection titles in the Results were revised to make them more informative.

Reviewer #3 (Remarks to the Author):

*A previous publication using glandless *Pgr cre/+ Foxa2 f/f* mice and gland-containing *Ltf-cre Foxa2 f/f* mice revealed critical role of *Foxa2* in pregnancy establishment. Based on previous study, the current study by Kelleher AM et al. confirmed that removal of the luminal epithelium is delayed and subsequent decidualization fails in LIF-replaced glandless but not gland-containing FOXA2-deficient mice. Importantly, this study newly found that adverse ripple effects of those dysregulated events in the endometrium lacking glands resulted in embryo resorption and pregnancy failure. The study is carefully designed and meticulously executed. The results of the manuscript would help understand the function of uterine glands in embryo implantation and decidualization. Although the current study includes some novel findings, the transcriptome analysis is confused and not well presented. There are several points to be addressed.*

Author Response: We appreciate the enthusiasm for our findings and the constructive comments. We have revised and improved the presentation and interpretation of the transcriptome analyses.

1. Abstract: Line 35-36: The authors do not provide any evidence of uterine gland dysfunction in women.

Author Response: The statement was removed from the Abstract.

2. Although the IHC analysis was examined in the luminal epithelium of the implantation chamber (crypt), the expression patterns of these proteins in Fig. 1b, Fig. 3a, and Suppl. Fig. 2a have been published by the previous PNAS paper.

Author Response: We have previously published the expression pattern of PTGS2 within implantation sites on GD 6 in LIF-repleted glandless and gland-containing mice (Kelleher et al, PNAS 2017), but not on subsequent days of pregnancy. Therefore, we chose to keep PTGS2 localization in the revised manuscript.

3. The mRNA-seq results are confused and descriptive. The analysis results do not provide clear biological meaning in the implantation process.

Author Response: The transcriptomic data and their presentation was reorganized and revised to address this comment in the revised manuscript. Importantly, the data provide a resource and foundation for future investigations into early pregnancy by our laboratory as well as others.

4. Fig. 2a and b: Why are the top 150 genes selected for the heatmap analysis? Are there any specific patterns of gene expression profiling in the top genes?

Author Response: All differentially expressed genes (DEGs) are now presented in Fig. 2a. We choose 150 genes to visually represent differences present between groups. Supplemental tables are provided with all DEGs, and raw data is deposited in the GEO.

5. How are "LE vs GE" genes selected from the previous data? Provide detailed method how two data sets are compared.

Author Response: Genes enriched in the GE were determined by comparing ($P < 0.05$, >2-fold) lists of genes expressed in isolated samples of LE and GE from our previous publication (Filant et al., 2013 *FASEB J*). The gene lists and analysis for that data can be found within that manuscript. Details of how the datasets were compared are provided in the revised manuscript (Lines 95-96).

6. The Venn diagram are confused. Where are 2303 genes and 361 genes in the Venn diagram (Line 109-111)?

Author Response: The numbers that were combined in the Venn diagrams to reach 2303 and 361 are now denoted with superscripts in the Venn Diagram for clarity and provided in the revised figure legend.

7. Suppl. Fig. 5: It is difficult to find the 2794 increased genes and the 2238 decreased genes the Venn diagram.

Author Response: We apologize for the discrepancy. The numbers are now updated in the revised text and agree with the figure and analysis.

8. Line 201-201: The authors mentions that glandless PgrCre/+Foxa2f/f mice have a defect of on-time implantation due to a delay in LE removal. When do PgrCre/+Foxa2f/f mice show LE removal during the pregnancy?

Author Response: There is a delay in the removal of the LE in glandless mice, but glandless mice displayed LE removal by 9 AM on GD 6 (Fig. 5a). Thus, the removal of the LE must occur between GD 5 night and GD 6 morning.

9. *The color scales of heatmap are opposite in Fig. 2 and Fig. 6. Keep the same pattern.*

Author Response: Thanks for pointing this out, the color scales are now the same in the revised manuscript.

REVIEWERS' COMMENTS:

Reviewer #1 (Remarks to the Author):

The authors have modified the manuscript and addressed the critiques satisfactorily.

Reviewer #2 (Remarks to the Author):

The authors have adequately addressed this reviewer's minor concerns.

Reviewer #3 (Remarks to the Author):

The authors have adequately addressed my concerns and the paper is improved accordingly.